# Activation mechanism of the insulin receptor revealed by cryo-EM structure of the fully liganded receptor–ligand complex

**Emiko Uchikawa[1†], Eunhee Choi[2†]\*, Guijun Shang[3], Hongtao Yu[2]\*, Xiao-chen Bai[1,4]\***

[1]Department of Biophysics, University of Texas Southwestern Medical Center, Dallas, United States; [2]Department of Pharmacology, Howard Hughes Medical Institute, University of Texas Southwestern Medical Center, Dallas, United States; [3]Department of Pharmacology, University of Texas Southwestern Medical Center, Dallas, United States; [4]Department of Cell Biology, University of Texas Southwestern Medical Center, Dallas, United States

**Abstract** Insulin signaling controls metabolic homeostasis. Here, we report the cryo-EM structure of full-length insulin receptor (IR) and insulin complex in the active state. This structure unexpectedly reveals that maximally four insulins can bind the 'T'-shaped IR dimer at four distinct sites related by 2-fold symmetry. Insulins 1 and 1' bind to sites 1 and 1', formed by L1 of one IR protomer and α-CT and FnIII-1 of the other. Insulins 2 and 2' bind to sites 2 and 2' on FnIII-1 of each protomer. Mutagenesis and cellular assays show that both sites 1 and 2 are required for optimal insulin binding and IR activation. We further identify a homotypic FnIII-2–FnIII-2 interaction in mediating the dimerization of membrane proximal domains in the active IR dimer. Our results indicate that binding of multiple insulins at two distinct types of sites disrupts the autoinhibited apo-IR dimer and stabilizes the active dimer.

DOI: https://doi.org/10.7554/eLife.48630.001

**\*For correspondence:**
Eunhee.Choi@UTSouthwestern.
edu (EC);
hongtao.yu@utsouthwestern.edu
(HY);
Xiaochen.Bai@UTSouthwestern.
edu (X-B)

[†]These authors contributed equally to this work

**Competing interests:** The authors declare that no competing interests exist.

## Introduction

Insulin receptor (IR) is a receptor tyrosine kinase (RTK) that plays essential roles in glucose metabolism and cell growth (*Ullrich et al., 1985*). Dysregulation of IR signaling is linked to many human diseases, such as diabetes and cancers (*Olefsky, 1976*). Unlike most RTKs, IR forms a stable homodimer covalently linked by multiple disulfide bonds (*Schäffer and Ljungqvist, 1992*; *Sparrow et al., 1997*). Previous mutagenesis and binding studies have suggested that IR has two distinct insulin-binding sites with different affinities for insulin, termed sites 1 and 2 (*De Meyts, 1994*). It has been further proposed that a single insulin molecule simultaneously engages site 1 of one IR promtomer and site 2 of the other, thus bridging the two IR protomers (*De Meyts, 1994*; *Schäffer, 1994*). A second insulin molecule binds to the equivalent, symmetry-related site in the IR dimer. These two insulins effectively crosslink the two IR protomers to activate IR.

The crystal structure of the extracellular domain of IR (comprising L1, CR, L2, FnIII-1,–2,–3 domains) in the absence of insulin was solved first (*McKern et al., 2006*), revealing an inverted 'V' shaped architecture. The crystal structure of the IR extracellular domain bound to insulin was subsequently determined (*Menting et al., 2013*), showing that one insulin molecule binds to the primary site, which consists of the L1 domain and the C-terminal helix of the α chain (α-CT). This primary site has been proposed to represent site 1. Recently, three groups, using single-particle electron

microscopy (EM), showed that insulin binding induces a large conformational change in IR (*Gutmann et al., 2018*; *Scapin et al., 2018*; *Weis et al., 2018*) and converts the overall architecture of IR dimer from an autoinhibited inverted 'V' shape to a 'T' shape. This conversion has been proposed to bring the two kinase domains of the IR dimer into close proximity to enable efficient trans-autophosphorylation and activation. Two of the three studies (*Gutmann et al., 2018*; *Weis et al., 2018*) also identified a new IR–insulin interface involving a loop of FnIII-1. This interface has been proposed to represent site 2 of IR. Thus, those structures revealed that the IR dimer binds to one or two insulins, with each insulin contacting the L1 domain of one protomer and α-CT and the FnIII-1 loop of the other. These findings seemingly support the 'crosslinking' model for IR activation, which posits that each insulin molecule simultaneously engages sites 1 and 2 of IR.

Here, we report the cryo-EM structure of the full-length human IR–insulin complex at an overall resolution of 3.2 Å. Unexpectedly, our structure reveals that each 'T'-shaped fully liganded IR dimer can bind maximally four insulins at four sites that are related by a two-fold symmetrical axis. Sites 1 and 1' are formed by the L1 domain of one IR protomer and α-CT and the FnIII-1 loop of the other. Sites 2 and 2' are located on the back of a main β sheet of the FnIII-1 domain of each protomer. Our mutagenesis, binding and cellular assays, along with insulin mutations in the previous literature (*De Meyts, 2015*; *Kristensen et al., 1997*), confirm this new IR–insulin interface as the authentic site 2. Therefore, our results revise and extend the widely accepted 'crosslinking' model of IR activation, and indicate that binding of multiple insulin molecules to two distinct types of sites on each IR dimer breaks the auto-inhibited state and fully activates IR.

## Results

### Overall structure of full-length human IR bound to insulin

We expressed and purified the full-length human insulin receptor (IR) using human HEK293F cells as the expression system (*Figure 1—figure supplement 1A,B*).), and reconstituted the IR–insulin complex in vitro. To improve the expression yield, we introduced several mutations in the intracellular domain of IR, including the D1120N mutation that inactivates its kinase activity and the Y960F mutation that has been shown to reduce clathrin-mediated endocytosis of IR (*White et al., 1988*; *Choi et al., 2019*). To exclude the possibility that the detergent micelles could affect insulin binding, we performed in vitro insulin binding assays. Recombinant IR isolated with detergent binds insulin with high affinity and displays strong negative cooperativity (Hill coefficients of ~0.62), comparable to our and other's cell-based insulin-binding results (*Figure 1—figure supplement 1C,D*) (*De Meyts et al., 1973*).

During cryo-EM image processing, initial 3D classification was produced without any symmetry imposed. All the resolved 3D reconstructions, however, exhibited almost perfect 2-fold symmetry (*Figure 1—figure supplement 2*), suggesting that insulins bind to the majority of IR in a symmetrical manner. The final refinement and reconstruction of the entire complex was thus performed with 2-fold symmetry applied, and was determined at an overall resolution of 3.2 Å (*Figure 1A*, and *Figure 1—figure supplement 3*, and *Table 1*).

To further improve the resolution, we performed a focused refinement (as described in our previous work, *Wong et al., 2014*) for the top part of the IR–insulin complex, including the L1, CR, L2, and FnIII-1 domains of IR as well as all the bound insulins. As a result, the resolution for the top part of the complex was improved to 3.1 Å (*Figure 1B*), allowing us to build a nearly complete atomic model for this region (*Figure 1B*, *Figure 1—figure supplements 2* and *4*). The cryo-EM densities for the FnIII-2 and FnIII-3 domains were less well resolved, presumably due to structural flexibility. Nevertheless, the crystal structures of FnIII-2 and FnIII-3 domains can be unambiguously docked into the cryo-EM density (*Figure 1A*), based on clear secondary structural features (*Figure 1—figure supplement 4A*). The cryo-EM density for the transmembrane domain (TM) can only be visualized after further 3D classification with local angular search (*Figure 1—figure supplements 2* and *4B*), but cannot be modeled due to the lack of side-chain densities. The densities of kinase domains were completely unresolved in the cryo-EM map.

In the full-length IR–insulin complex, the IR dimer exists in an extended 'T'-shaped arrangement with 2-fold symmetry (*Figure 1*), in agreement with the 2D class averages of full-length IR in the presence of insulin obtained previously by negative-stain EM (*Gutmann et al., 2018*) as well as the

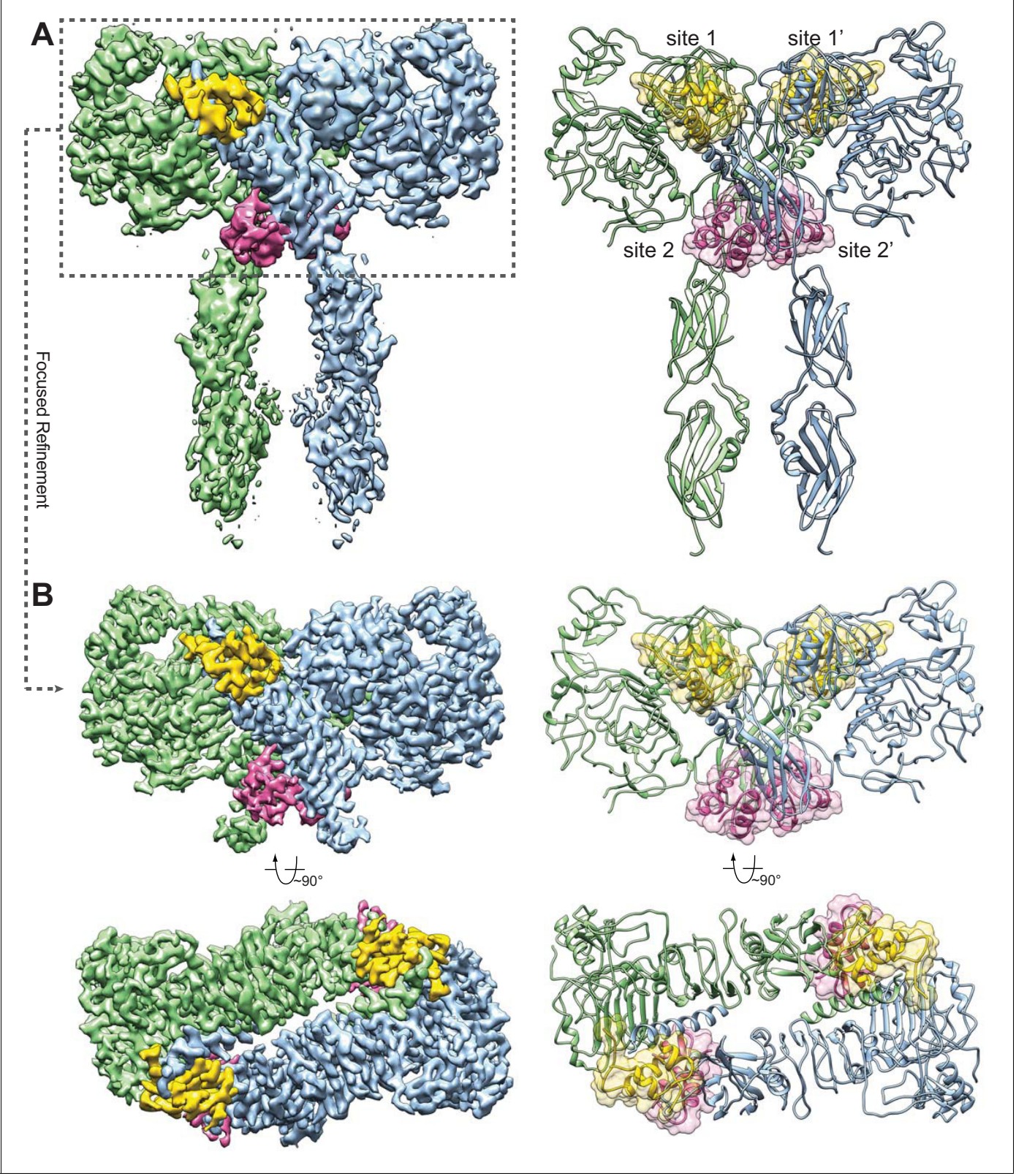

**Figure 1.** Overall structure of the IR–insulin complex. (**A**) 3D reconstruction of the IR dimer with four insulins bound (left) and the corresponding ribbon representation of this complex (right). (**B**) 3D reconstruction of IR dimer with four insulins bound after forced 3D refinement (left) and the corresponding ribbon representation (right).

DOI: https://doi.org/10.7554/eLife.48630.002

*Figure 1 continued on next page*

*Figure 1 continued*

The following figure supplements are available for figure 1:

**Figure supplement 1.** Purification of the full-length human insulin receptor (IR).
DOI: https://doi.org/10.7554/eLife.48630.003
**Figure supplement 2.** Flowchart of cryo-EM data processing.
DOI: https://doi.org/10.7554/eLife.48630.004
**Figure supplement 3.** Cryo-EM analysis of the IR–insulin complex.
DOI: https://doi.org/10.7554/eLife.48630.005
**Figure supplement 4.** Cryo-EM density of the IR–insulin complex.
DOI: https://doi.org/10.7554/eLife.48630.006
**Figure supplement 5.** Detecting variations in insulin occupancies of IR by focused classification.
DOI: https://doi.org/10.7554/eLife.48630.007

recently solved cryo-EM structure of the IR extracellular domain bound to insulin (*Scapin et al., 2018*). The horizontal part of the 'T' is composed of the L1, CR and L2 domains of both IR promoters. The vertical piece of the 'T' consists exclusively of the FnIII-1, FnIII-2 and FnIII-3 domains of the IR dimer.

The apo-form of the IR dimer exhibits an inverted 'V' shape, with the two FnIII-3 domains separated by ~120 Å (*McKern et al., 2006*). Because the FnIII-3 domain is connected to the transmembrane (TM) domain and intracellular kinase domains through a short linker (*Figure 1—figure supplement 1B*), the two kinase domains are likely separated by a long distance and cannot undergo efficient trans-autophosphorylation in this autoinhibited conformation. The distance between the two FnIII-3 domains in the 'T'-shaped insulin-bound IR dimer, however, is reduced to ~40 Å. In addition, the partially resolved density for the single-pass TM domain suggests that the

**Table 1.** Cryo-EM data collection and model statistics.

|  | IR-insulin complex |
| --- | --- |
| **Data collection and processing** |  |
| Magnification | 46,730 |
| Voltage (kV) | 300 |
| Electron exposure (e⁻/Å²) | 50 |
| Defocus range (μm) | 1.5 – 3 |
| Pixel size (Å) | 1.07 |
| Final particle number | *235,707* |
| Map resolution (Å) | 3.1 |
| Map Sharpening $B$ factor | −100 |
| **Model Refinement** |  |
| **Rms deviations** |  |
| Bonds (Å) | 0.007 |
| Angels (°) | 0.885 |
| **Validation** |  |
| MolProbity score | 1.66 |
| Clashscore | 3.36 |
| Rotamer outliers (%) | 0 |
| **Ramachandran plot** |  |
| Favored (%) | 90.52 |
| Allowed (%) | 9.33 |
| Outliers (%) | 0.15 |

DOI: https://doi.org/10.7554/eLife.48630.008

two TM domains may dimerize in the IR–insulin complex (*Figure 1—figure supplement 4B*). The two intracellular kinase domains are thus expected to be in close proximity in this 'T'-shaped conformation, enabling their trans-autophosphorylation. Therefore, this conformation of the IR dimer represents an active form.

Strikingly, we can identify four insulins bound to four sites in the fully liganded IR dimer (*Figure 1*). Because of the 2-fold symmetry, there are two distinct types of insulin and insulin-binding sites in the complex, denoted as insulins 1, 1', 2, and 2'; and sites 1, 1', 2, and 2'. For simplicity, we will only describe insulins 1 and 2, which bind to sites 1 and 2 on one side of the 'T'-shaped dimer. Insulin 1 at site 1 and insulin 2 at site 2 bury 706 and 394 Å$^2$ of solvent-exposed surface areas of IR, respectively. Thus, site 1 likely has higher affinity in insulin binding than site 2.

Insulin has a molecular weight of ~6 kDa, which is much smaller than that of IR. The initial global 3D classification with coarse angular sampling may not be able to identify subsets of particles that exhibit partially liganded IR conformations. To separate the particles with different numbers of bound ligands, we applied our previously developed approaches—symmetry expansion in combination with focused classification on insulin with signal subtraction of the receptor (*Bai et al., 2015*; *Zhou et al., 2015*). As the insulin molecules are too small to be aligned on their own, during the focused classification, the orientations of the insulins were kept fixed as those determined by the C2 refinement of the entire complex. As a result, we can identify two major classes having different stoichiometries for insulin binding (*Figure 1—figure supplement 5*). In particular, 63% of the particles have four insulins bound (two each at sites 1 and 2), while 33% of the particles have three insulins bound (two at site 1; one at site 2) (*Figure 1—figure supplement 5B*). There is only a subtle difference between the structures of IR with three or four insulins bound, which indicates that binding of insulins to both sites 2 and 2' is not required for the maintenance of the T-shaped IR dimer (*Figure 1—figure supplement 5A*). Importantly, the vast majority (96%) of the T-shaped particles have at least one insulin bound to site 2 or 2', suggesting that insulin binding to site 2 is important for IR to adopt the active T shape.

During the revision of this manuscript, another study deposited in the preprint server bioRxiv also showed a low-resolution cryo-EM map of the IR extracellular domain with four insulins bound at sites 1 and 2 (*Gutmann et al., 2019*). The 3:1 or 4:1 stoichiometry of insulin binding to the IR dimer and the two types of binding sites are consistent with previous IR–insulin binding assays that indicated the coexistence of high- and low-affinity insulin-binding sites (*Schäffer, 1994*). Those results were, however, interpreted with a binding equilibrium of 2:1 stoichiometry, and in light of our findings here may need to be reexamined.

## A tripartite interface between insulin 1 and site 1 stabilizes the active IR dimer

Insulin 1 binds mainly through the primary site formed by the L1 domain and α-CT (*Figure 2A*, left panel). Based on our cryo-EM map, we cannot distinguish whether the α-CT is from the same protomer or the neighboring one, as the density of the linker between L1 domain and α-CT is completely missing in our structure. Nevertheless, based on previous biochemical and structural results (*Chan et al., 2007*), IR is an intertwined domain-swapped homodimer. Thus, the two elements that constitute this primary site, the L1 domain and α-CT, are from two different IR protomers. This primary insulin–IR interface is nearly identical to that observed in the crystal structure of the IR extracellular domain bound to insulin (*Menting et al., 2013*), and will not be described in detail. To confirm that insulin binding to this site is essential for IR activation, we introduced R14E (*Whittaker and Whittaker, 2005*) and F714A mutations into IR to disrupt this interface, and showed that both IR R14E and F714A mutants, when expressed in human 293 cells, exhibited deficient insulin-dependent IR activation (*Figure 2B, and 2C*).

In addition to this primary interface, insulin 1 makes contact with a loop of the FnIII-1 domain from the IR protomer that has donated α-CT (*Figure 2A*, left panel). This secondary interface was first observed in the previously reported cryo-EM study of the IR extracellular domain bound to insulin (*Scapin et al., 2018*), but could not be accurately modeled due to the low-resolution density map and had been erroneously assigned as site 2. Our significantly improved cryo-EM map allowed the delineation of this new interface in near-atomic detail. To distinguish this interface from the authentic site 2 described in the next section, we propose to name the primary interface as site 1a and this new secondary interface as site 1b (*Figure 2A*, left panel).

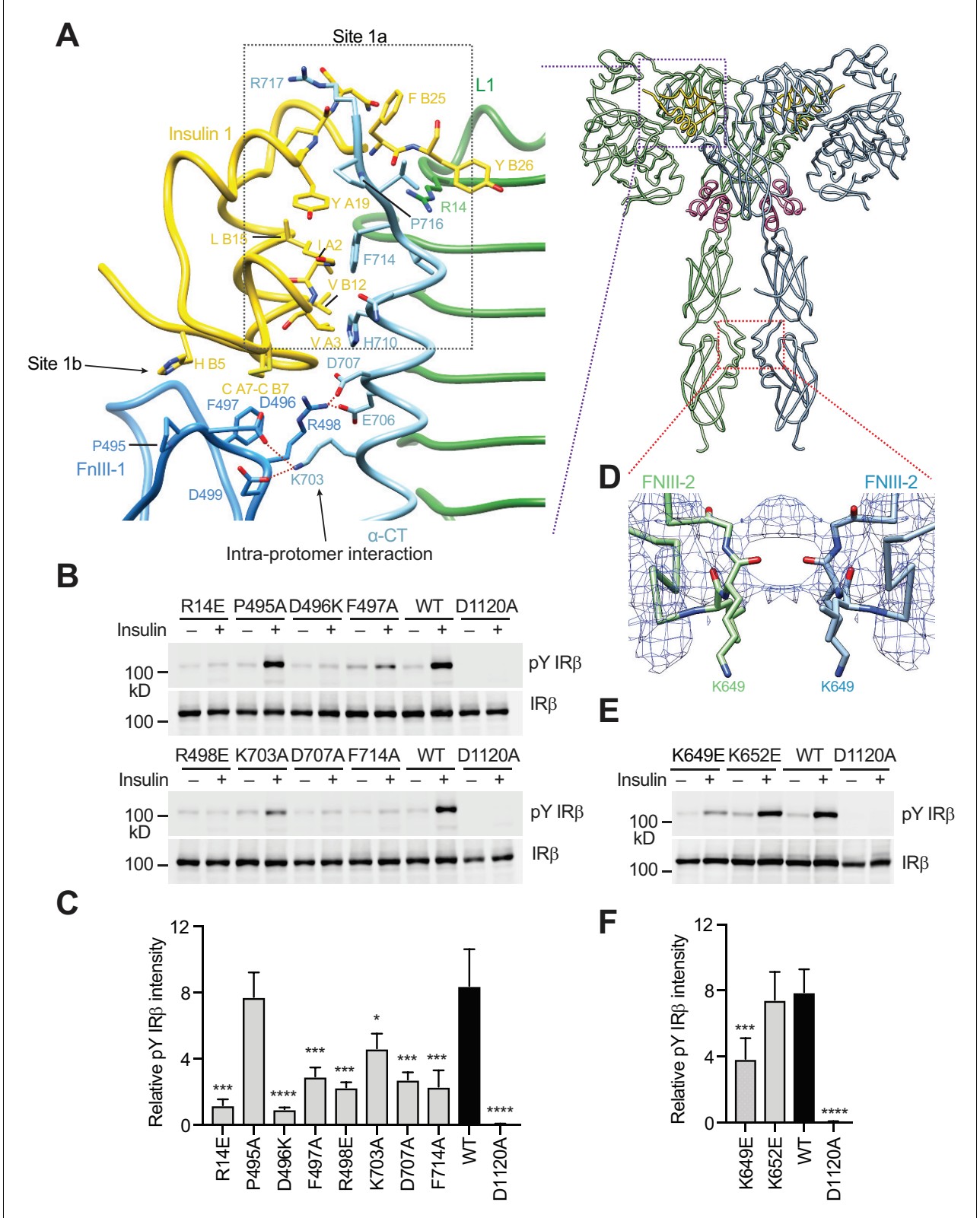

**Figure 2.** A tripartite interface between insulin one and IR site one stabilizes the active IR dimer. (**A**) Close-up view of the tripartite interaction between insulin 1 and IR L1, α-CT, and FnIII-1 domains (left panel) and ribbon diagram of the IR–insulin complex (right panel). The location of this interaction in the IR active dimer is indicated by a purple box in the right panel. (**B**) Insulin-induced IR autophosphorylation in 293FT cells expressing IR wild-type (WT) or indicated point mutants that were expected to lose the tripartite interaction. The IR kinase-dead mutant (D1120A) was used as the negative control.

*Figure 2 continued on next page*

*Figure 2 continued*

(C) Quantification of the western blot data shown in B) (Mean ± SD). Each experiment was repeated three times. Significance calculated using two-tailed students *t*-test; between WT and mutants; *p<0.05, ***p<0.001, and ****p<0.0001. (D) Close-up view of the homotypic interaction between two adjacent FnIII-2 domains. The location of this interaction in the IR active dimer is indicated by a red box in A) (right panel). (E) Insulin-induced IR autophosphorylation in 293FT cells expressing IR WT and the indicated mutants. The kinase dead mutant (D1120A) was used as the negative control. (F) Quantification of the western blot data shown in E) (Mean ± SD). Each experiment was repeated four times. Significance calculated using two-tailed students *t*-test; between WT and mutants; ***p<0.001 and ****p<0.0001.

DOI: https://doi.org/10.7554/eLife.48630.009

At site 1b, two disulfide-bonded cysteines (Cys A7 and Cys B7) and His B5 of insulin pack against Pro495, Phe497, and Arg498 in the IR FnIII-1 domain through hydrophobic interactions (*Figure 2A*, left panel). In addition, insulin binding induces the formation of a new interface between the FnIII-1 domain and α-CT of the same protomer (*Figure 2A*, left panel), which only exists in the 'T'-shaped active dimer. This interface involves a cluster of charged residues, including Asp496, Arg498 and Asp499 in the FnIII-1 domain and Lys703, Glu706 and Asp707 in α-CT, which form multiple salt bridges (*Figure 2A*, left panel). Notably, an IR D707A mutation has been found in patients with leprechaunism, a rare genetic disease with severe insulin signaling defects (*Hart et al., 1996*). This mutation may affect IR activation by disrupting this interface.

To test the functional relevance of site 1b, we mutated the residue Phe497 of IR (which contacts insulin) to alanine, and tested its effect on IR activation. The F497A mutant showed markedly reduced insulin-dependent IR activation (*Figure 2B, and 2C*). We next introduced several point mutations (D496A, R498E, K703A, and D707A) into IR, which were designed to break the interaction between α-CT and the FnIII-1 domain. The D496A, R498E, D707A (the disease-causing mutation), and K703A mutants all showed greatly reduced activation (*Figure 2B, and 2C*). These results validate the functional importance of the interactions at site 1b in IR activation.

Taken together, binding of insulin 1 to IR site 1 involves a tripartite interface: insulin–L1/α-CT (site 1a), insulin–FnIII-1 loop (site 1b), and α-CT–FnIII-1 loop (site 1b). Thus, insulin 1 simultaneously engages site 1a and 1b to bridge the two IR protomers. We propose that this strong tripartite interaction is critical for the formation of the 'T'-shaped active IR dimer by driving the rearrangement of the two IR protomers. Conceptually, the binding mode of insulin 1 is reminiscent of the 'crosslinking' model proposed previously, except that this insulin crosslinks two sub-sites of site 1, as opposed to sites 1 and 2.

## Insulin 2 binds to a novel binding site on the FnIII-1 domain of IR

We have identified a novel insulin-binding site for insulin 2 binding, which is located on the back side of a major β sheet of the FnIII-1 domain (*Figure 3A, and 3C*). Residues from both A and B chains of insulin 2 contribute to this interaction (*Figure 3B*). These residues are located on the side of insulin opposite to that involved in the binding to site 1, and largely overlap with those in the insulin hexamer interface (*Table 2*) (*Weiss, 2009*). Several basic residues from IR FnIII-1, including Arg479, Lys484, Arg488, and Arg554, participate in this interaction (*Figure 3B*). In addition, hydrophobic residues from IR FnIII-1, such as Leu486, Leu552, and Pro537, pack tightly against residues Leu A13, Tyr A14, Leu A16, Leu B6, Ala B14, Leu B17, and Val B18 in insulin (*Figure 3B*, and *Table 2*). Many site 2 residues are conserved in vertebrate IR proteins, but some are not conserved in the IGF1R (*Figure 3—figure supplement 1*). We superimposed the model of receptor-bound insulin 2 onto that of free insulin, which was previously determined by X-ray crystallography (PDB: 1MSO) (*Smith et al., 2003*). This comparison revealed no major structural differences between the free and bound insulin.

Most of the insulin residues involved in the binding to site 2, which include Ile A10, Ser A12, Leu A13, Glu A17, His B10, Glu B13, and Leu B17, have been correctly mapped by alanine scanning mutagenesis in the previous work (*De Meyts, 2015*; *Kristensen et al., 1997*), and these insulin site 2 mutations markedly reduced receptor-binding affinity (*De Meyts, 2015*; *Kristensen et al., 1997*). Importantly, most of these previously defined site 2 residues of insulin do not contact IR at site 1b. Thus, the previous study (*Scapin et al., 2018*) has incorrectly assigned site 1b as site 2. Our structure of the IR–insulin complex has revealed the authentic site 2 of IR–insulin binding and represents the first structure of the fully liganded, active IR.

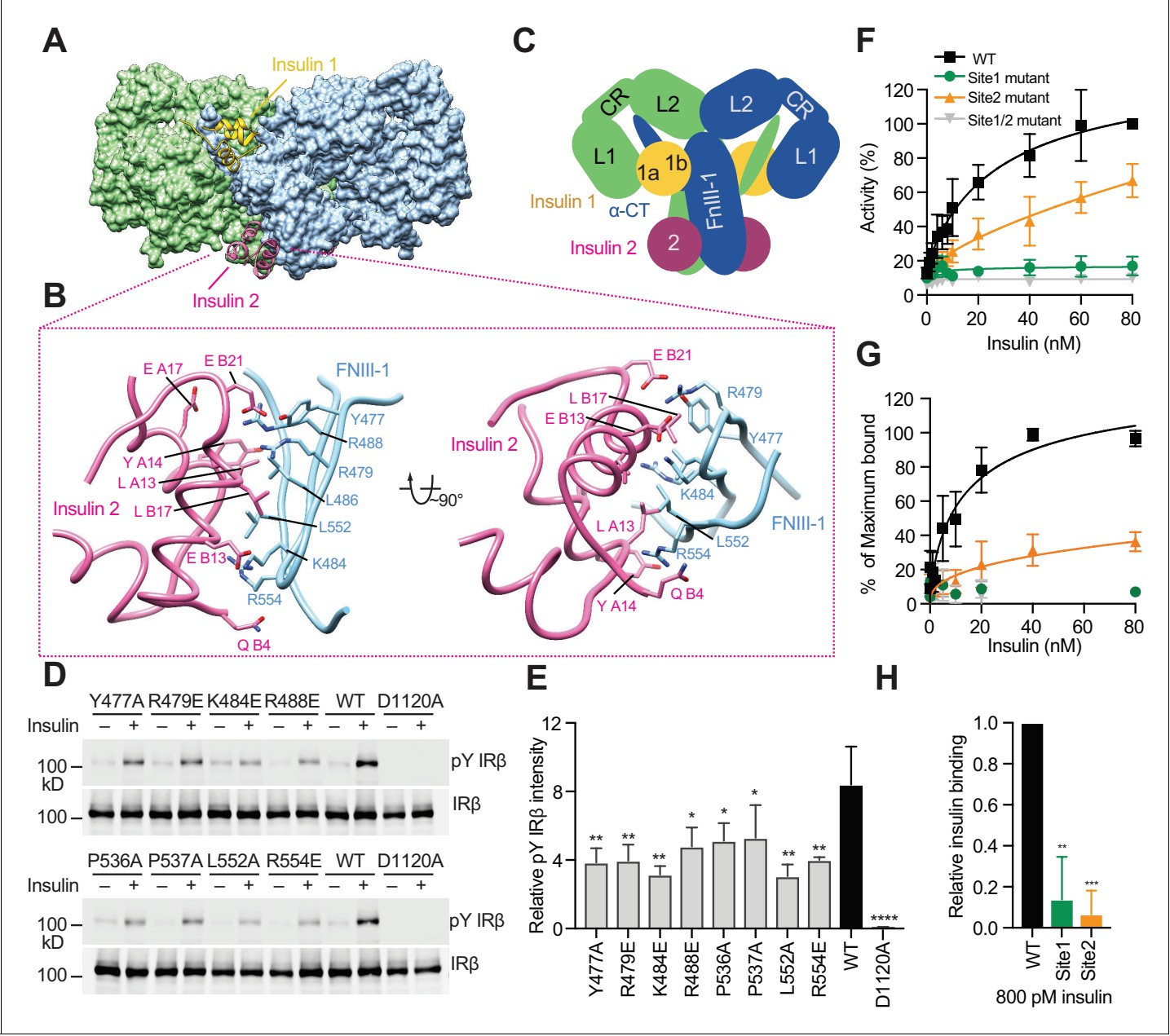

**Figure 3.** Insulin 2 binds to a novel binding site on the FnIII-1 domain of IR. (A) Overall view of the two distinct insulin-binding sites. (B) Close-up view of insulin-binding site 2. (C) Cartoon representation of the IR–insulin complex depicting the two distinct types of insulin-binding sites (total of 4 sites per IR dimer related by 2-fold symmetry). (D) Insulin-induced IR autophosphorylation in 293FT cells expressing IR wild-type (WT) or the indicated site 2 mutants, assessed by quantitative western blotting with a phospho-tyrosine (pY) IRβ antibody. Kinase dead mutant (D1120A) was used as a negative control. Expression levels of IRβ are monitored by anti-Myc blotting against the C-terminal Myc-tag. (E) Quantification of the western blot data shown in D (Mean ± SD). Each experiment was repeated three times. Significance calculated using two-tailed students $t$-test; between WT and mutants, *$p<0.05$, **$p<0.01$, and ****$p<0.0001$. (F) Dose-responsive curves of the insulin-stimulated autophosphorylation of wild-type (WT) IR or IR mutants (Mean ± SD). Site 1 mutant, R14E; site 2 mutant K484E/L552A; site 1/2 mutant, R14E/K484E/L552A. Significance calculated using One-way ANOVA with Brown-Forsythe test; between WT and mutants, $p<0.0001$. See *Figure 3—figure supplement 2A*. (G) Binding of insulin labeled with Alexa Fluor 488 to 293FT cells expressing IR WT or IR mutants (Mean ± SD). Significance calculated using One-way ANOVA with Brown-Forsythe test; between WT and site 2 mutant, $p<0.0001$. Symbols are the same as in F. (H) Binding of insulin labeled with Alexa Fluor 488 to 293FT cells expressing IR WT or IR mutants at 800 pM concentration of insulin. Significance calculated using two-tailed students $t$-test; between WT and mutants, **$p<0.01$, and ***$p<0.001$.
DOI: https://doi.org/10.7554/eLife.48630.010

The following figure supplements are available for figure 3:

**Figure supplement 1.** Sequence alignment of IR proteins from human (Hs), mouse (Mm), xenopus (Xl), and Hs IGF1R.

*Figure 3 continued on next page*

*Figure 3 continued*
DOI: https://doi.org/10.7554/eLife.48630.011
**Figure supplement 2.** Dose dependency of IR activation by insulin.
DOI: https://doi.org/10.7554/eLife.48630.012
**Figure supplement 3.** Cellular localization of IR WT and mutants.
DOI: https://doi.org/10.7554/eLife.48630.013

Next, we introduced multiple mutations in IR (Y477A, R479E, K484E, R488E, P536A, P537A, L552A, and R554E) designed to weaken this insulin 2-binding interface. Consistent with our structure-based predictions, all these IR mutants showed decreased insulin-dependent activation (*Figure 3D, and 3E*). Among them, the K484E and L552A mutants were the most defective. In a dose-response assay, the IR K484E/L552A double mutant (site 2 mutant) was defective in insulin-dependent activation in a wide range of insulin concentrations (*Figure 3F* and *Figure 3—figure supplement 2A*), although the effect of this site 2 mutation was not as severe as that of the site 1 R14E mutation. Furthermore, both cell-based and in vitro insulin-binding assays showed that the IR site 2 K484E/L552A mutant indeed exhibited much weaker insulin-binding affinity, as compared to wild-type IR, particularly at the physiological concentrations of insulin (*Figure 3G and H*, *Figure 1—figure supplement 1C*, *Figure 3—figure supplement 2B and C*). Moreover, all site 2 IR mutants can localize to the plasma membrane, similar to wild-type IR (*Figure 3—figure supplement 3*), suggesting that the introduced mutations are unlikely to induce protein mis-localization or misfolding. These results validate the importance of insulin 2 binding to site 2 of IR in insulin-dependent activation of IR.

## Hinge motions of IR protomers upon insulin binding

The structures of the L1/CR segment and the FnIII-1/-2/-3 segment of insulin-bound IR can be superimposed perfectly onto those of apo-IR, suggesting that no domain rotations occur in these two structural segments during insulin binding. However, the overall structure of the IR protomer with insulin bound is more compacted, as a result of two large hinge motions around the L2 domain (*Figure 4A, B and C*). Such rigid-body rotations have been previously described (*Scapin et al., 2018*), but the structural details of these rearrangement cannot be elucidated due to the moderate resolution and incompleteness of the active IR dimer in that study.

A large conformational change takes place between the CR and L2 domains, with the loop connecting CR and L2 (residues 302–310) as the hinge (*Figure 4C*). This hinge loop has different conformations in apo- and insulin-bound IR (*Figure 4D*). In particular, Glu287 and Lys310 form a salt bridge to stabilize the insulin-bound loop conformation. Along with the conformational change of the hinge loop, the L1/CR domains of IR rotate ~90° toward the L2 domain upon insulin binding, creating two new inter-domain contacts (*Figure 4B, and 4C*). The first contact is formed by Glu697 and Phe701 in the N-terminal portion of α-CT and Arg345 and Gly346 on the lateral surface of the β sheet in the L2 domain (*Figure 4B*). As a result, α-CT can simultaneously engage both L1 and L2 domains of the other IR protomer, tethering these two domains together. Another new contact is formed mainly through the interaction between a loop (residues 85–90) of the L1 domain and the first α-helix (residues 323–331) of the L2 domain.

Another hinge motion upon insulin binding occurs around the linker between the L2 and FnIII-1 domains, resulting in a new inter-domain interaction between residues 454–460 of L2 and residues 572–578 in a loop of FnIII-1. Consequently, FnIII-1,-2, and -3 domains as a rigid body rotate away from the L2 domain by ~50° (*Figure 4C*). Through the combination of these two hinge motions, the

**Table 2.** Summary of the insulin and IR residues involved in site 2 binding.

| Insulin residues | IR residues |
|---|---|
| Ile A10, Ser A12, Leu A13, Tyr A14, Leu A16, Glu A17, Gln B4, Leu B6, His B10, Glu B13, Ala B14, Leu B17, Val B18, Glu B21 | Tyr477, Arg479, Ser481, Asp483, Lys484, Leu486, Arg488, Asp535, Pro537, Asn547, Pro549, Gly550, Trp551, Leu552 |

DOI: https://doi.org/10.7554/eLife.48630.014

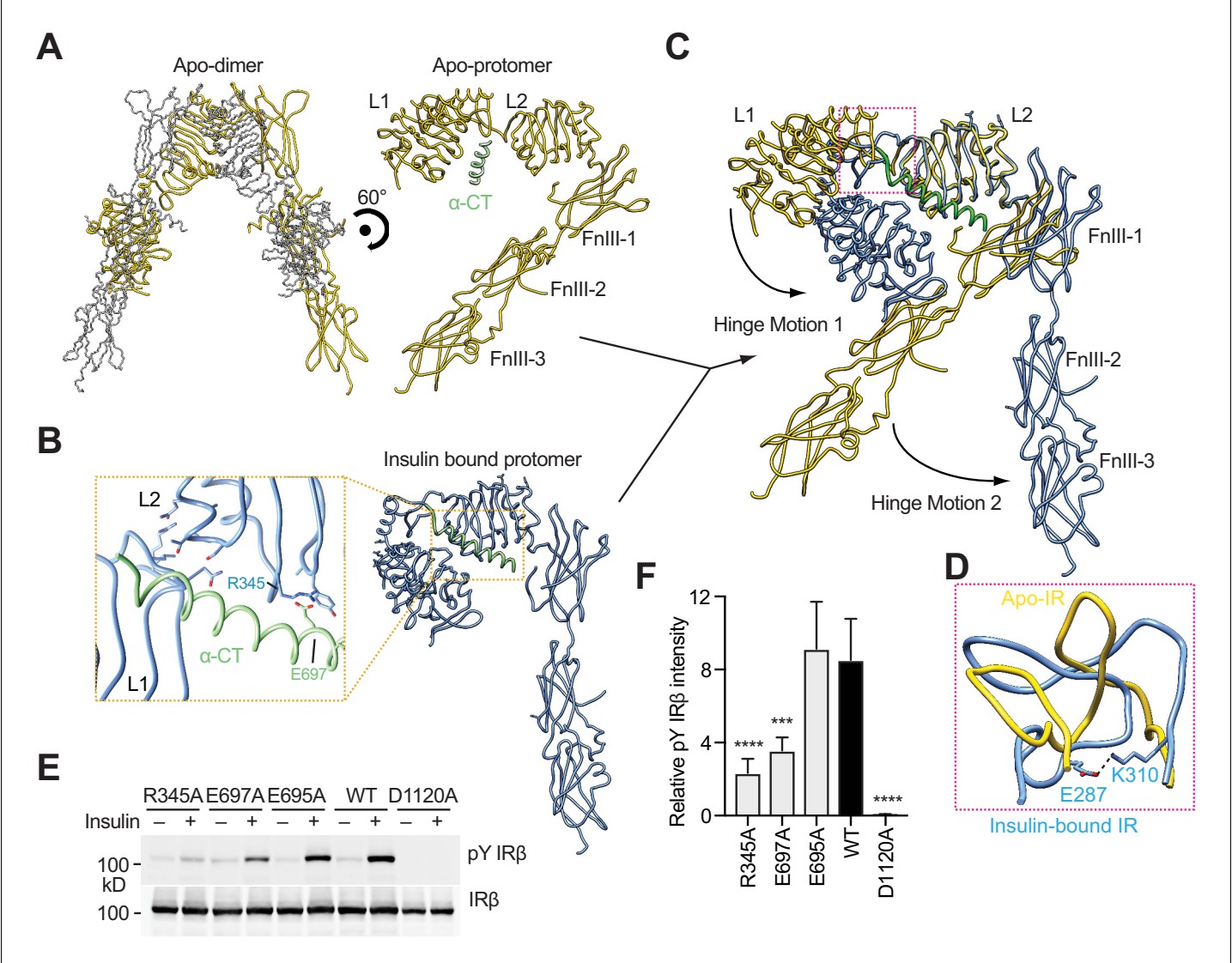

**Figure 4.** Hinge motions of IR protomers upon insulin binding. (A) Overall view of apo-IR dimer and protomer. (B) Overall view of IR protomer with insulinbound in a more compacted active conformation. Inset (yellow box), close-up view of the inter-domain interactions that stabilize the active conformation. (C) Superposition between insulin-free IR protomer (yellow) and insulin-bound IR protomer (blue) by aligning the L2 domain, revealing two different hinge motions. (D) Superposition of the loop connecting CR and L2 (residues 302–310) of the insulin-free IR protomer (yellow) onto that of insulin-bound IR protomer (blue). The location of this loop in the IR protomer is indicated by a pink box in C. (E) Insulin-induced IR autophosphorylation in 293FT cells expressing wild-type (WT) IR or indicated mutants. The kinase dead mutant (D1120A) was used as a negative control. (F) Quantification of the western blot data shown in E (Mean ± SD). Each experiment was repeated five times. Significance calculated using two-tailed students *t*-test; between WT and mutants, ***p<0.001 and ****p<0.0001.

DOI: https://doi.org/10.7554/eLife.48630.015

overall architecture of the IR protomer is converted an inverted 'J' shape (*Figure 4B*). Two such promoters make the 'T' shaped active dimer.

The α-CT domain plays a critical role in stabilizing the active conformation of the IR protomer. In particular, Glu697 of α-CT forms a salt bridge with Arg345 in the L2 domain (*Figure 4B*). Both residues are far away from the bound insulins. Mutations of Glu697 and Arg345 to alanine diminished IR activation by insulin (*Figure 4E, and 4F*). In contrast, mutation of a neighboring residue Glu695 in α-CT that was not involved in inter-domain interactions had no effect (*Figure 4E, and 4F*). These results confirm the biological relevance of the observed active conformation of the IR protomer.

Finally, we have identified a homotypic interaction between the FnIII-2 domains of the two active IR protomers. The FnIII-2–FnIII-2 interaction is mediated by residues 647–653 in an extended loop (*Figure 2D*). Due to the lack of the side chain densities, the details of this interaction are unclear. This interaction is likely to be functionally important, however, as mutation of Lys649, which is located in the center of this dimer interface, diminished IR activation, whereas mutation of Lys652, which is at the periphery of this interface, had no effect (*Figure 2E, and 2F*). Similar homotypic interactions between membrane-proximal domains have been found in other RTKs (*Yuzawa et al., 2007*). These interactions are thought to precisely position the two stalk domains at the cell membrane and help to bring the two intracellular kinase domains into proximity for trans-autophosphorylation.

## Mechanism of IR activation by multi-site insulin binding

Our mutagenesis results have shown that optimal IR activation requires the binding of multiple insulins to both sites 1 and 2 (*Figure 3F and G*, *Figure 1—figure supplement 1C* and *Figure 3—figure supplement 2*). In addition, our 3D classification results further suggest that binding of at least one insulin to the two site 2 s is required for the formation of the 'T'-shaped dimer. According to the cryo-EM structure, formation of the active IR dimer is mainly driven by the tripartite interface formed by insulin 1, L1, α-CT, and FnIII-1 at site 1. This raises the interesting question why insulin 2 promotes IR activation.

Previous studies have suggested that insulin binding to site 1 of apo-IR or IGF-1R can release the autoinhibited conformation, which is an important step during IR or IGF-1R activation (*Kavran et al., 2014*; *Xu et al., 2018*). We superimposed the L1 domain of insulin-bound IR, along with insulin 1, onto that of apo-IR. Insulin 1 bound at site 1a sterically clashes with the linker between the FnIII-1 and FnIII-2 domains in apo-IR (*Figure 5*). Thus, insulin 1 binding at site 1a will disrupt the inter-apo-protomer interaction 1 between L1 and FnIII-2 domains, which is important in maintaining the autoinhibited IR dimer.

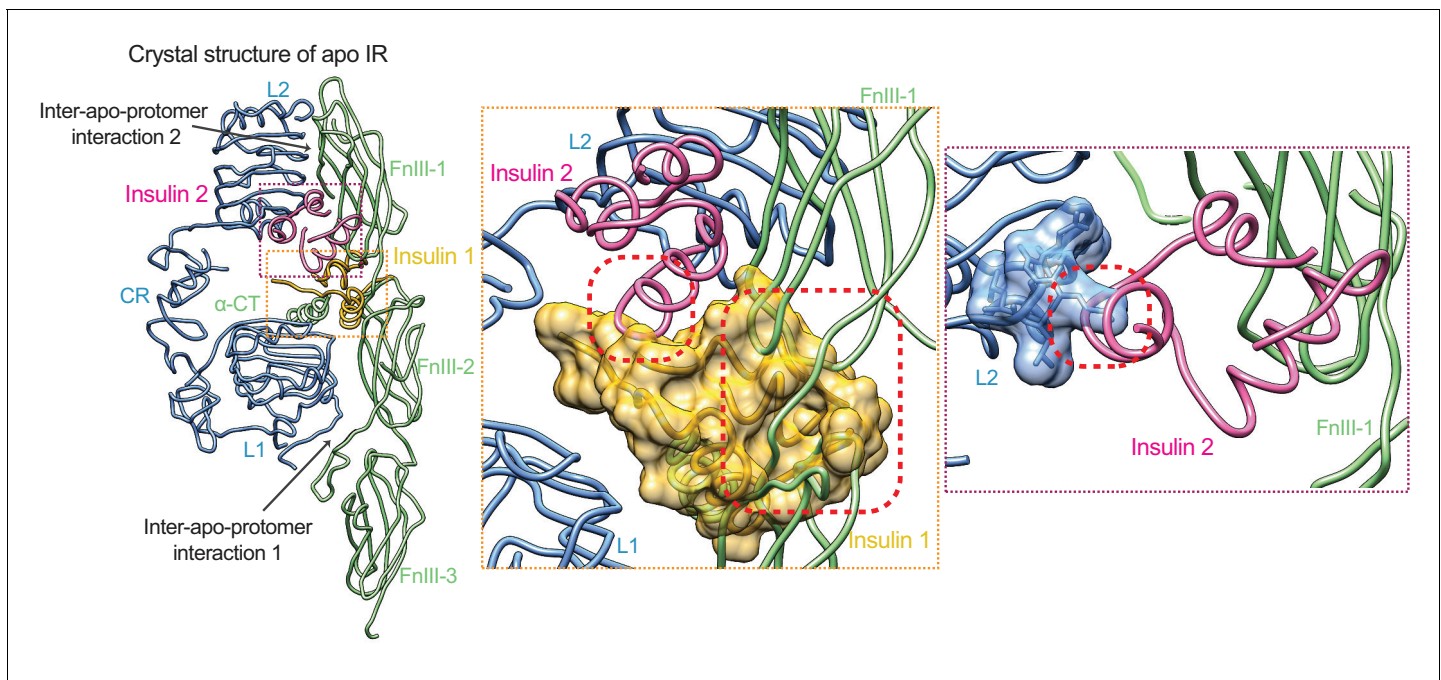

**Figure 5.** Activation mechanism of IR. The crystal structure of apo-IR dimer (PDB: 4ZXB) superimposed with insulins 1 and 2, by aligning the insulin-free and insulin 1-bound L1 domains, and the insulin-free and insulin 2-bound FnIII-1 domain, respectively. Half of the apo-dimer is omitted for clarity. Upper inset (purple box), insulin 2 binding at apo-IR sterically clashes with the opposite protomer. Lower inset (yellow box), insulin 1 binding at apo-IR sterically clashes with the opposite protomer, as well as insulin 1. The clash sites are indicated by dashed red boxes.
DOI: https://doi.org/10.7554/eLife.48630.016

Interestingly, the crystal structure of IGF-1R with IGF1 bound (*Xu et al., 2018*) shows that IGF1 binding to site 1a separates the L1 domain of one protomer from the FnIII-2 domain of the other protomer. However, due to the long flexible linker between the CR and L2 domains, the inter-apo-protomer interaction 2 between the L2 and FnIII-1 domains on the top part of the inverted 'V'-shaped dimer still exists, even with IGF bound at site1a. This result indicates that one IGF molecule cannot completely break the autoinhibited conformation of IGF-1R. By analogy, one insulin molecule is likely to be insufficient to break the autoinhibited IR dimer.

We next superimposed insulin 2 and the FnIII-1 domain of insulin-bound IR dimer onto that of apo-IR dimer. Thr A8 and Ser A9 of insulin 2 bound at site 2 clashes with a β-turn motif in the L2 domain of the apo-IR dimer. Thus, insulin binding to site 2 will destabilize the inter-apo-protomer interaction 2 between the L2 domain of one protomer and the FnIII-1 domain of the other (*Figure 5*). Furthermore, the two insulins bound to sites 1a and 2 in the apo-IR clash with each other (*Figure 5*). These additional steric clashes, generated by the simultaneous binding of two insulins to sites 1a and 2, may further drive the two apo-IR protomers apart and completely destabilize the autoinhibited IR dimer (*Figure 6A*). This could explain why the full activation of IR requires the binding of both insulins 1 and 2.

Once the autoinhibited IR dimer is disrupted by insulin binding, each insulin protomer is free to undergo two hinge motions, forming an inverted 'J'-shaped protomer (*Figure 6A*). Based on the current models, it is still unclear how the two 'J'-shaped protomers form the 'T'-shaped dimer. This process has to involve coordinated structural rearrangements and inter-protomer rotations. The 'T'-shaped active dimer is predominantly stabilized by a tripartite interface between insulin 1, L1, α-CT, and FnIII-1 domains, without the involvement of insulin 2. Moreover, the membrane proximal regions

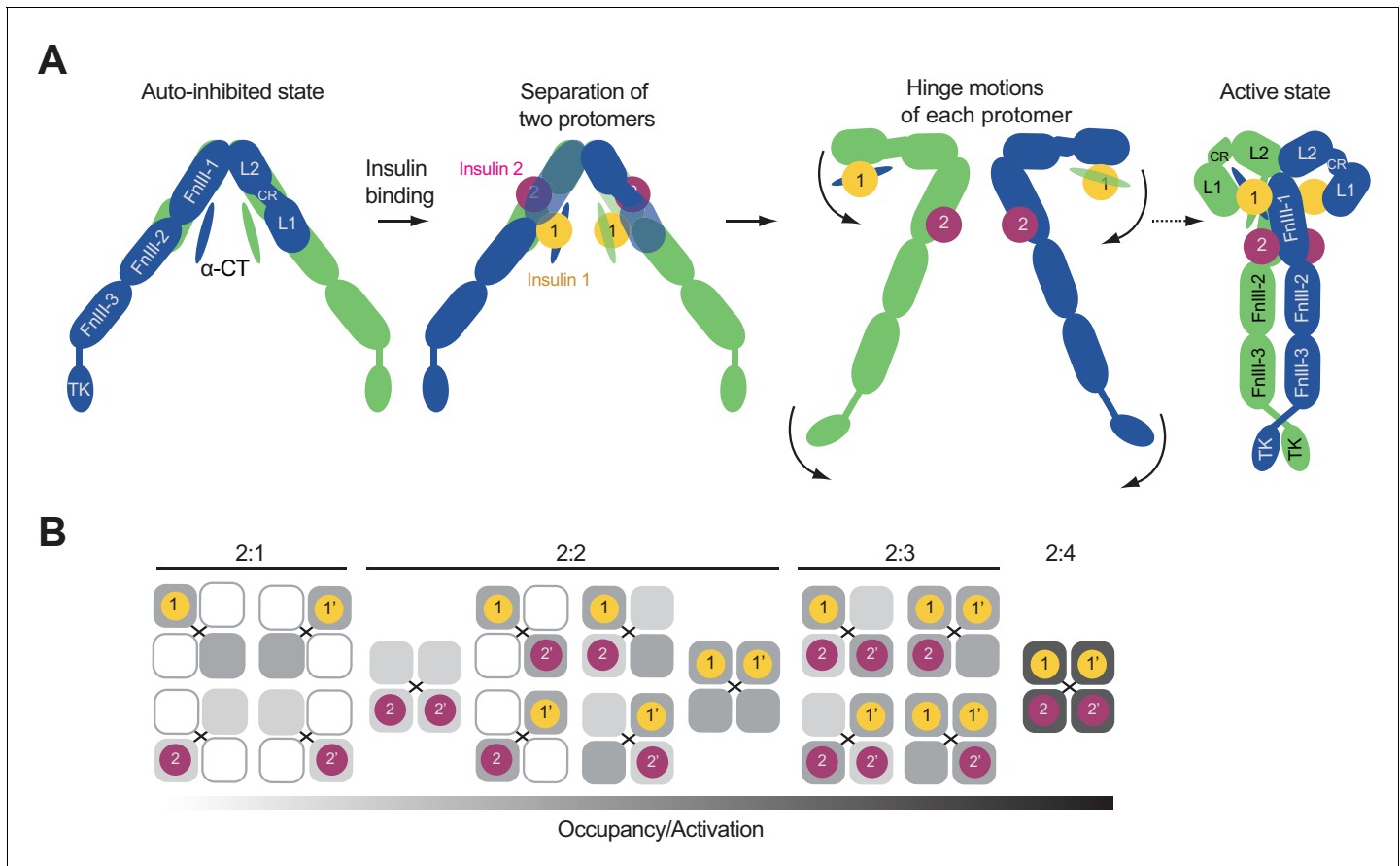

**Figure 6.** Proposed working model of insulin induced activation of IR. (**A**) Cartoon representation of a working model for insulin-induced IR activation. (**B**) Binding of IR and insulin in a concentration-dependent manner. Insulin binding at IR site 1 and site 2 is required for IR activation.

DOI: https://doi.org/10.7554/eLife.48630.017

from both protomers is brought together by a homotypic interaction between the two FnIII-2 domains. Finally, these structural rearrangements promote the lateral TM-TM interaction, and enable trans-autophosphorylation between the two intracellular kinase domains (*Figure 6A*). An endocytosis pathway, which involves the binding of MAD2-BUBR1 as well as the clathrin adaptor protein AP2 to the IR intracellular domain, then triggers the internalization of the active IR dimer and stops insulin signaling at the plasma membrane (*Choi et al., 2016*).

## Discussion

Our structural and functional analyses indicate that binding of maximally four insulins to two distinct types of sites disrupt the autoinhibited IR dimer. The relaxed IR dimer with insulin bound then undergo structural rearrangements to form a 'T'-shaped active dimer, which is stabilized by a tripartite interaction between insulin 1, L1, α-CT, and FnIII-1 domains. This binding mode strongly supports the previous insulin 'cross-linking' model. Additionally, we observed that the 'T'-shaped IR active dimer is further stabilized the homotypic interaction between the two FnIII-2 domains from the two protomers. This specific conformation of IR dimer facilitates the dimerization of the TM domain, thereby bringing the two kinase domains into close proximity for trans-autophosphorylation. Our study provides key mechanistic insight into the initiation of insulin signaling and paves the road for the eventual therapeutic intervention of diseases caused by insulin dysregulation.

*De Meyts et al. (1973)* have proposed that the binding of multiple insulins to IR occurs with negative cooperativity. The 2:3 or 2:4 stoichiometry of insulin binding to the IR and the two types of binding sites are consistent with previous IR–insulin binding assays that indicated the coexistence of high- and low-affinity insulin-binding sites (*Schäffer, 1994*). To understand the cooperativity between the binding of multiple insulins to different sites, we obtained a dose-response curve of insulin-dependent activation of IR WT and the site 2 mutant (*Figure 3F*). Both dose-response curves of IR WT and site 2 mutant could be fitted with the Hill equation, with Hill coefficients of ~0.64 and ~0.42, respectively, which are indicative of negative cooperativity in insulin binding to IR. Similar results were obtained by in vitro insulin binding assays (*Figure 1—figure supplement 1C*). Because the IR site 2 mutant exhibits even stronger negative cooperativity and because insulin binding to site 2 is required for maximum insulin binding and full activation of IR, our results suggest that insulin binding to site 2 might dampen the anti-cooperativity between insulin binding to site 1 and site 1'. The combination of cooperative binding between sites 1 and 2 and anti-cooperativity binding between sites 1 and 1' likely underlies the complicated shapes of the binding curves between IR and insulin. This unique system allows IR to respond differently to a wide range of insulin concentrations that various tissues experience at different metabolic states (*Figure 6B*).

Our functional analyses indicate that site 1 and site 2 are both required for optimal IR activation. It remains unknown which site is occupied first. Based on the different binding affinities between these two sites and all the existing structural models, we speculate that one insulin binding to site 1 happens first, but may only lead to a partially active IR with an asymmetrical conformation. The subsequent binding of one insulin to site 2 facilitates the binding of the second insulin 1 to site 1', and thereby driving the formation of the symmetrical dimer and full IR activation. This hypothesis is partially supported by our new structural model that binding of two site-1 insulins and at least one site-2 insulin is necessary to promote the formation of the 'T'-shaped IR dimer. Solving the structures of IR site 1 and 2 mutants is needed to further test this hypothesis.

We also do not know for certain how many insulins are required for IR activation. It is possible that, at low insulin concentrations, IR with partial insulin occupancy (e.g. one insulin at site 1) can achieve asymmetric conformations and exhibit different levels of activation (*Weis et al., 2018*) (*Figure 6B*). In addition, insulin can trigger both the metabolic and mitogenic responses through IR (*De Meyts, 2000*). It is tempting to speculate that the conformational differences of IR caused by different stoichiometries of bound insulins may provide a mechanism in defining insulin signaling specificity, which could potentially allow the same ligand (i.e. insulin) to generate distinct signaling outputs from the same receptor (i.e. IR). Future biophysical and in vivo experiments, preferably those involving single-molecule techniques, are required to define the intermediate steps of IR activation. Our structure of the fully liganded IR-insulin complex serves as an important guide for designing these experiments.

# Materials and methods

## Key resources table

| Reagent type (species) or resource | Designation | Source or reference | Identifiers | Additional information |
|---|---|---|---|---|
| Strain, strain background (*Escherichia coli*) | One Shot Stbl3 Chemically Competent *E .coli* | Life Technologies | C7373-03 | |
| Strain, strain background (*Escherichia coli*) | DH10Bac bacteria | Thermo Fisher | 10361012 | |
| Cell line (*Homo-sapiens*) | 293FT, Human embryonic kidney | Invitrogen | R70007 | |
| Cell line (*Homo-sapiens*) | HEK293 GnTI–, Human embryonic kidney | ATCC | CRL-3022 | |
| Cell line (*Homo-sapiens*) | HEK293, Human embryonic kidney | ATCC | CRL-1573 | |
| Transfected construct (*Homo-sapiens*) | pCS2-human Insulin receptor WT-MYC | *Choi et al., 2016* | | transfected construct (human) |
| Transfected construct (*Homo-sapiens*) | pCS2-human Insulin receptor R14E-MYC | This paper | | Materials and methods, subsection: Insulin receptor activation assay, In vivo insulin-binding assay |
| Transfected construct (*Homo-sapiens*) | pCS2-human Insulin receptor P495A-MYC | This paper | | Materials and methods, subsection: Insulin receptor activation assay |
| Transfected construct (*Homo-sapiens*) | pCS2-human Insulin receptor D496K-MYC | This paper | | Materials and methods, subsection: Insulin receptor activation assay |
| Transfected construct (*Homo-sapiens*) | pCS2-human Insulin receptor F497A-MYC | This paper | | Materials and methods, subsection: Insulin receptor activation assay |
| Transfected construct (*Homo-sapiens*) | pCS2-human Insulin receptor D1120A-MYC | This paper | | Materials and methods, subsection: Insulin receptor activation assay |
| Transfected construct (*Homo-sapiens*) | pCS2-human Insulin receptor R498E-MYC | This paper | | Materials and methods, subsection: Insulin receptor activation assay |
| Transfected construct (*Homo-sapiens*) | pCS2-human Insulin receptor K703A-MYC | This paper | | Materials and methods, subsection: Insulin receptor activation assay |
| Transfected construct (*Homo-sapiens*) | pCS2-human Insulin receptor F714A-MYC | This paper | | Materials and methods, subsection: Insulin receptor activation assay |
| Transfected construct (*Homo-sapiens*) | pCS2-human Insulin receptor K649E-MYC | This paper | | Materials and methods, subsection: Insulin receptor activation assay |
| Transfected construct (*Homo-sapiens*) | pCS2-human Insulin receptor K652E-MYC | This paper | | Materials and methods, subsection: Insulin receptor activation assay |
| Transfected construct (*Homo-sapiens*) | pCS2-human Insulin receptor Y477A-MYC | This paper | | Materials and methods, subsection: Insulin receptor activation assay |
| Transfected construct (*Homo-sapiens*) | pCS2-human Insulin receptor R479E-MYC | This paper | | Materials and methods, subsection: Insulin receptor activation assay |
| Transfected construct (*Homo-sapiens*) | pCS2-human Insulin receptor K484E-MYC | This paper | | Materials and methods, subsection: Insulin receptor activation assay |

*Continued on next page*

*Continued*

| Reagent type (species) or resource | Designation | Source or reference | Identifiers | Additional information |
|---|---|---|---|---|
| Transfected construct (*Homo-sapiens*) | pCS2-human Insulin receptor R488E-MYC | This paper | | Materials and methods, subsection: Insulin receptor activation assay |
| Transfected construct (*Homo-sapiens*) | pCS2-human Insulin receptor P536A-MYC | This paper | | Materials and methods, subsection: Insulin receptor activation assay |
| Transfected construct (*Homo-sapiens*) | pCS2-human Insulin receptor P537A-MYC | This paper | | Materials and methods, subsection: Insulin receptor activation assay |
| Transfected construct (*Homo-sapiens*) | pCS2-human Insulin receptor L552A-MYC | This paper | | Materials and methods, subsection: Insulin receptor activation assay |
| Transfected construct (*Homo-sapiens*) | pCS2-human Insulin receptor R554E-MYC | This paper | | Materials and methods, subsection: Insulin receptor activation assay |
| Transfected construct (*Homo-sapiens*) | pCS2-human Insulin receptor K484E,L552A-MYC | This paper | | Materials and methods, subsection: Insulin receptor activation assay, In vivo insulin-binding assay |
| Transfected construct (*Homo-sapiens*) | pCS2-human Insulin receptor R345A-MYC | This paper | | Materials and methods, subsection: Insulin receptor activation assay |
| Transfected construct (*Homo-sapiens*) | pCS2-human Insulin receptor E697A-MYC | This paper | | Materials and methods, subsection: Insulin receptor activation assay |
| Transfected construct (*Homo-sapiens*) | pCS2-human Insulin receptor E695A-MYC | This paper | | Materials and methods, subsection: Insulin receptor activation assay |
| Transfected construct (*Homo-sapiens*) | pCS2-human Insulin receptor R14E,K484E,L552A-MYC | This paper | | Materials and methods, subsection: Insulin receptor activation assay, In vivo insulin-binding assay |
| Transfected construct (*Homo-sapiens*) | pCS2-human Insulin recetor Y960F, S962A, R1333A, I1334A, L1335A, L1337A-Myc | *Choi et al., 2016* | | transfected construct (human) |
| Transfected construct (*Homo-sapiens*) | pEZT-BM-human Insulin receptor (Y960F, S962A, D1120N, R1333A, I1334A, L1335A, L1337A)—3C-Tsi3-His | This paper | | Materials and methods, subsection: Insulin receptor activation assay, In vivo insulin-binding assay |
| Transfected construct (*Homo-sapiens*) | pEZT-BM-human Insulin receptor (K484E, L552A, Y960F, S962A, D1120N, R1333A, I1334A, L1335A, L1337A)—3C-Tsi3-His | This paper | | Materials and methods, subsection: Insulin receptor activation assay, In vivo insulin-binding assay |
| Antibody | Rabbit monoclonal anti-IR-pY1150/1151 | Cell signaling | 19H7 | WB (1:1000) |
| Antibody | Mouse monoclonal anti-MYC | Roche | Clone 9E10 | WB (1:1000), IF (1:500) |
| Antibody | Anti-rabbit IgG (H+L) (Dylight 800 conjugates) | Cell signaling | 5151 | WB (1:5000) |
| Antibody | Anti-mouse IgG (H+L) (Dylight 680 conjugates) | Cell signaling | 5470 | WB (1:5000) |
| Commercial assay or kit | Micro BCA Protein Assay Kit | Thermo Scientific | 23235 | |
| Commercial assay or kit | Alexa Fluor 488 | Thermo Scientific | A10235 | |

*Continued on next page*

*Continued*

| Reagent type (species) or resource | Designation | Source or reference | Identifiers | Additional information |
|---|---|---|---|---|
| Commercial assay or kit | Q5 site directed mutagenesis kit | NEB | E0554S | |
| Commercial assay or kit | Gibson Assembly Master Mix | NEB | E2611 | |
| Chemical compound, drug | cOmplete Protease Inhibitor Cocktail | Roche | 5056489001 | |
| Chemical compound, drug | PhosSTOP | Roche | 4906837001 | |
| Chemical compound, drug | BMS536924 | Tocris | 4774 | |
| Software, algorithm | CytExpert 2.1 | Beckman Coulter | | https://www.beckman.com/flow-cytometry/instruments/cytoflex/software |
| Software, algorithm | Protparam | ExPASy | | https://web.expasy.org/protparam/ |
| Software, algorithm | Prism 7 | GraphPad | N/A | |
| Other | Sepharose 4B resin | GE Healthcare | 17012001 | |
| Other | Superose 6 Increase 10/300 GL | GE Healthcare | 29091596 | |
| Other | Superdex 75 increase 10/300 GL | GE Healthcare | 29148721 | |
| Other | Dynabeads Protein G | Invitrogen | 10003D | |
| Other | DMEM, high glucose | Thermo Fisher | 11965–118 | |
| Other | Pierce NHS-Activated Agarose, Dry | Thermo Fisher Scientific | 26197 | |
| Other | Amicon Ultra-15 Centrifugal Filter Units | Milliporesigma | UFC9100 | |
| Other | Amicon Ultra-0.5 Centrifugal Filter Units | Milliporesigma | UFC5100 | |
| Other | Quantifoil R 1.2/1.3 grid Au300 | quantifoil | Q37572 | |
| Other | ProLong Gold Antifade reagent with DAPI | Invitrogen | P36935 | |
| Other | Cellfectin | Invitrogen | 10362100 | |
| Other | Lipofectamine 2000 | Invitrogen | 11668019 | |
| Other | Human Insulin | Sigma | I2643 | |

## Protein expression and purification

The amino acid numbering for IR starts after the signal peptide. For the activation assay, the short isoform of human insulin receptor (IR) was cloned into pCS2-MYC as described previously (*Choi et al., 2016*). For structural studies, a cDNA fragment encoding human IR with seven mutations (Y960F, S962A, D1120N, R1333A, I1334A, L1335A, L1337A), followed by the Human Rhinovirus 3C recognition site, affinity purification tag Tsi3 (T6SS secreted immunity protein three from *Pseudomonas aeruginosa*) and His$_8$ tag, was cloned into the pEZT-BM vector (*Morales-Perez et al., 2016*). The mutations were designed to prevent IR endocytosis.

This plasmid was transformed into the DH10Bac bacteria (Thermo Fisher, 10361012) for production of bacmid DNA. Recombinant baculovirus was produced by transfecting Sf9 cells with the bacmid DNA using Cellfectin (Cellfectin Reagent, Invitrogen). Protein was expressed with suspension-adapted HEK293 GnTI$^-$ cells by infecting the virus at a ratio 1:10 (virus: cell, v/v). The infected cells

were supplemented with 6 mM sodium butyrate to increase protein expression. The cells were cultured for 60 hr at 37°C and 8% $CO_2$.

IR was firstly purified by affinity purification using the strong binding between the T6SS effector Tse3 and its immunity protein Tsi3 (*Lu et al., 2014*) and further polished by gel filtration chromatography. Briefly, Cells were harvested, resuspended in buffer A [20 mM Hepes pH 7.5, 400 mM NaCl, 5% glycerol and one tablet of cOmplete Protease Inhibitor Cocktail (Roche)] and lysed by sonication. Lysate was centrifuged for 60 min at 45,000 g to pellet the membranes. The pellet containing the membrane fraction was resuspended in the buffer A containing 1% n-Dodecyl-β-D-Maltopyranoside (DDM) and 0.1% Cholesteryl Hemisuccinate (CHS) (D310-CH210 Anatrace). After incubation at 4°C for 3 hr on the bottle roller, the insoluble parts were removed by centrifugation for 60 min at 45,000 g. The supernatant was supplemented with 2 mM $CaCl_2$ and loaded on to the Tse3-conjugated sepharose 4B resin (GE heathcare). The resin was washed with buffer A containing 2 mM $CaCl_2$, 0.03% DDM, and 0.003% CHS. The HRV3C protease was added to the resin in the washing buffer and incubated for 12 hr at 4°C. Untagged IR was eluted from the resin and applied to the size exclusion chromatography (Superose 6 Increase 10/300 GL GE Health care) in the final buffer [20 mM Hepes pH 7.5, 200 mM NaCl, 0.03% DDM, and 0.003% CHS]. The peak fractions were mixed with human insulin (I2643, Sigma) in a 1:2 molar ratio and concentrated with Amicon Ultra concentrator 100 kD cut off (Millipore) to approximately 7 mg/ml. two additional molar equivalents of insulin were added to the final sample to saturate the binding.

## EM data acquisition

The cryo-EM grid was prepared by applying 3 µl of protein samples to glow-discharged Quantifoil R1.2/1.3 300-mesh gold holey carbon grids (Quantifoil, Micro Tools GmbH, Germany). Grids were blotted for 5.0 s under 100% humidity at 4°C before being plunged into liquid ethane using a Mark IV Vitrobot (FEI). Micrographs were acquired on a Titan Krios microscope (FEI) operated at 300 kV with a K2 Summit direct electron detector (Gatan), using a slit width of 20 eV on a GIF-Quantum energy filter. EPU software (FEI) was used for automated data collection following standard FEI procedures. A calibrated magnification of 46,730X was used for imaging, yielding a pixel size of 1.07 Å on images. The defocus range was set from −1.5 µm to −3 µm. Each micrograph was dose-fractionated to 30 frames under a dose rate of 4 e⁻/pixel/s, with a total exposure time of 15 s, resulting in a total dose of about 50 e⁻/Å$^2$.

## Image processing

The detailed image processing statistics are summarized in *Figure 1—figure supplements 2* and *3*, and *Table 1*. Motion correction was performed using the MotionCorr2 program (*Zheng et al., 2017*), and the CTF parameters of the micrographs were estimated using the GCTF program (*Zhang, 2016*). Initially,~2000 particles were picked with EMAN2 from a few micrographs (*Tang et al., 2007*). All other steps of image processing were performed using RELION (*Scheres, 2012a*; *Scheres, 2012b*). Class averages representing projections in different orientations selected from the initial 2D classification were used as templates for automatic particle picking from the full datasets. Extracted particles were binned three times and subjected to 2D classification. Particles from the classes with fine structural features were selected for 3D classification. Approximately 20,000 particles were selected to generate the initial model in RELION. Particles from the 3D classes showing good secondary structural features were selected and re-extracted into the original pixel size of 1.07 Å. 3D refinements with C2 symmetry imposed resulted in 3D reconstructions to 3.2 Å resolution. To improve the resolution, we performed another round of 3D refinement with a soft mask around the top part of complex, leading to improved resolution for this region. Moreover, we classified the refined particles set by using local search in combination with small angular sampling, resulting a new class showing improved density for TM domain.

To separate the particles with different insulin occupancies, we performed focused classification with symmetry expansion and signal subtraction, as described in our previous papers (*Bai et al., 2015*; *Zhou et al., 2015*). We duplicated each particle and rotated the second copy of each particle around the two fold symmetrical axis for 180° to put the site 1/2 and site 1'/2' in the same position for focused classicization. The symmetry expanded particles set was classified with a local mask around site 1 or two without any angular alignment. As IR is much larger than insulin, the density

corresponding to IR was subtracted from each experimental particle for consistent comparison. The resulting classes show similar level of intensity for insulin 1, but different levels of intensities for insulin 2. The C1 refinement was performed for particles from the classes having weak insulin two density.

All resolutions were estimated by applying a soft mask around the protein density using the gold-standard Fourier shell correlation (FSC) = 0.143 criterion (*Scheres and Chen, 2012c*).

## Model building and refinement

Model building of the IR–insulin complex was initiated by docking each individual domain derived from the crystal structures of IR (PDB ID: 4ZXB; *Croll et al., 2016*) and insulin (PDB ID: 1MSO; *Smith et al., 2003*) into the high-resolution map from the focused refinement in the program Chimera (*Pettersen et al., 2004*). The model was manually adjusted in Coot (*Emsley et al., 2010*) and refined against the map by using the real space refinement module with secondary structure and non-crystallographic symmetry restraints in the Phenix package (*Adams et al., 2010*). To check for potential model overfitting, the coordinates in the model were refined against one of the half maps calculated from half of the dataset in RELION. The FSC curve from this model and map was only slightly worse than that calculated from the refined model and the summed map, suggesting no overfitting. The crystal structures of FnIII-2/3 domains were rigid-body fitted into the cryo-EM density in Chimera (*Pettersen et al., 2004*), showing good agreement. Model geometries were assessed by using Molprobity as a part of the Phenix validation tools and summarized in *Table 1*.

## Insulin receptor activation assay

293FT cells (R70007, Invitrogen) were cultured in high-glucose DMEM supplemented with 10% (v/v) FBS, 2 mM L-glutamine, and 1% penicillin/streptomycin. Cells were free from mycoplasma contamination. Plasmid transfection was performed with LipofectamineTM 2000 (Invitrogen). After 1 day, the cells were serum starved for 14 hr and treated with 10 nM of human insulin (I2643, Sigma) or the indicated concentrations for 5 min. The cells were washed with cold PBS and incubated with the cell lysis buffer [50 mM Hepes pH 7.4, 150 mM NaCl, 10% (v/v) Glycerol, 1% (v/v) Triton X-100, 1 mM EDTA, 100 mM sodium fluoride, 2 mM sodium orthovanadate, 20 mM sodium pyrophosphate, 0.5 mM dithiothreitol (DTT), 2 mM phenylmethylsulfonyl fluoride (PMSF)] supplemented with cOmplete Protease Inhibitor Cocktail (Roche) and PhosSTOP (Roche) on ice for 1 hr. After centrifugation at 20,817 g at 4°C for 20 min, the concentrations of cell lysate were measured using Micro BCA Protein Assay Kit (Thermo Scientific). Cell lysates (50 μg total proteins) were analyzed by SDS-PAGE and Western blotting. Anti-IR-pY1150/1151 (19H7, Cell signaling; labeled as pY IRβ) and anti-MYC (9E10, Roche; labeled as IRβ) were used as primary antibodies. For quantitative Western blots, anti-rabbit immunoglobulin G (IgG) (H+L) (Dylight 800 conjugates) and anti-mouse IgG (H+L) (Dylight 680 conjugates) (Cell signaling) were used as secondary antibodies. The membranes were scanned with the Odyssey Infrared Imaging System (LI-COR, Lincoln, NE). Levels of pY IRβ were normalized to total IRβ levels and shown as intensities relative to that of IR WT in untreated cells.

## In vivo insulin-binding assay

To conjugate insulin with Alexa Fluor 488 (A10235, Thermo Scientific), human insulin (I2643, Sigma) was dissolved in 20 mM Hepes pH 7.5, 100 mM NaCl, and 2 mM EDTA, and Alexa Fluor 488 NHS Ester was dissolved in dimethylsulfoxide (DMSO). Insulin and the dye were mixed in a molar ratio of 1:2 and incubated for 12 hr at 4°C. Labeled Insulin was separated from the free dye by size exclusion chromatography (Superdex 75 increase 10/300 GL, GE Healthcare) in the buffer containing 20 mM Hepes pH 7.5, 100 mM NaCl, and 2 mM EDTA. The insulin-binding assay was performed as described previously with slight modifications (*Murphy et al., 1982*). Briefly, plasmid transfection was performed with LipofectamineTM 2000 into 293FT cells. After 1 day, the cells were serum starved for 14 hr. To inhibit insulin receptor endocytosis, the plates were then changed to serum free DMEM containing the IR/IGF1R inhibitor BMS536924 (1 μM; Tocris) for an additional 1 hr before insulin treatment. The medium was replaced with serum-free DMEM containing Alexa Fluor 488-labeled insulin at the indicated concentrations. After 5 min of incubation in $CO_2$ incubators at 37°C, the medium was removed and the plates were washed with cold PBS. Cells were dissociated in 0.5 mM EDTA in PBS at 37°C for 5 min, transferred into conical tubes, centrifuged for 5 min at 200 g at

4°C. After two more washes with cold PBS, the cells were resuspended in 300 µl of cold PBS and transferred into tubes for FACS analysis. Samples were kept on ice and analyzed within 30 min. The samples were analyzed on a Cytoflex Flow Cytometer (Beckman Coulter). At least 10,000 cells were analyzed for each sample. Mean fluorescence intensities were calculated with the software CytExpert 2.1 (Beckman Coulter). Nonspecific binding was measured in untransfected control cells and subtracted from the data.

### In vitro insulin-binding assay

For in vitro insulin-binding assay, cDNA fragments encoding human IR with seven mutations (Y960F, S962A, D1120N, R1333A, I1334A, L1335A, L1337A; insulin binding site wild-type) and nine mutations (K484E, L552A, Y960F, S962A, D1120N, R1333A, I1334A, L1335A, L1337A; insulin binding site two mutant), followed by the human Rhinovirus 3C recognition site, affinity purification tag Tsi3 and $His_8$ tag, were cloned into the pEZT-BM vector. Protein was expressed with suspension-adapted HEK293 cells as described in Protein expression and purification.

40 µg of anti-IRβ antibody (sc-57342; Santa Cruz Biotechnology) was added to 12 mg of Dynabeads Protein G (Invitrogen) in 400 µl PBS, and incubated for 1 hr at room temperature. After a single wash, the beads were resuspended in 400 µl of binding buffer [20 mM Hepes pH 7.5, 200 mM NaCl, 0.03% DDM, and 0.003% CHS]. The indicated amount of IR Alexa Fluor 488 labeled insulin, and 100 nM of BSA were incubated in 200 µl of binding buffer containing cOmplete protease inhibitor Cocktail (Roche) for 2 hr on ice. 10 µl beads were added and incubated on a rotator for 30 min at room temperature. The beads were washed two times with the binding buffer. The bound proteins were eluted with 50 µl of binding buffer containing 2% SDS for 10 min at 50°C. The samples were diluted with 150 µl of binding buffer. The fluorescence intensities were measured in a microplate reader (CLARI0star; BMG LABTECH). Nonspecific binding was measured in samples of Alexa Fluor 488 labeled insulin and 100 nM of BSA without IR, and subtracted from the data. The samples were analyzed by SDS-PAGE and Western blotting. IR concentration was estimated using molecular weight 305,276.92 Da in the dimeric receptor as calculated by Protparam (ExPASy).

### Immunofluorescence

Indirect immunofluorescence microscopy was performed on cells grown on chamber slides and fixed in cold methanol at −20°C for 10 min. The fixed cells were washed with cold acetone twice and incubated with PBS for 30 min and blocking buffer [3% BSA and 0.1% (v/v) TritonX-100 in PBS] for 1 hr, and then treated with diluted antibodies in blocking buffer at 4°C overnight. After being washed, cells were incubated with fluorescent secondary antibodies and mounted in ProLong Gold Antifade reagent with DAPI (Invitrogen). Images were acquired and analyzed as previously described (*Choi et al., 2016*).

### Quantification and statistical analysis

The number of independent experiment, the method used in statistical test, and the statistical significance are indicated in each figure legend and source manuscript files.

## Acknowledgements

Single particle cryo-EM data were collected at the University of Texas Southwestern Medical Center (UTSW) Cryo-Electron Microscopy Facility that is funded by a Cancer Prevention and Research Institute of Texas (CPRIT) Core Facility Support Award (RP170644). We thank D Nicastro and D Stoddard for facility access and data acquisition. XB is the Virginia Murchison Linthicum Scholar in Medical Research at UTSW. Research in his lab is supported in part by CPRIT (RR160082) and the Welch foundation (I-1944–20180324). HY is an Investigator with the Howard Hughes Medical Institute, and supported by grants from CPRIT (RP120717-P2 and RP160667-P2) and the Welch Foundation (I-1441).

## Additional information

### Funding

| Funder | Grant reference number | Author |
|---|---|---|
| Cancer Prevention and Research Institute of Texas | RR160082 | Xiao-chen Bai |
| Welch Foundation | I-1944 | Xiao-chen Bai |
| Cancer Prevention and Research Institute of Texas | RP120717-P2 | Hongtao Yu |
| Cancer Prevention and Research Institute of Texas | RP160667-P2 | Hongtao Yu |
| Howard Hughes Medical Institute | | Hongtao Yu |
| Welch Foundation | I-1441 | Hongtao Yu |

The funders had no role in study design, data collection and interpretation, or the decision to submit the work for publication.

### Author contributions

Emiko Uchikawa, Eunhee Choi, Data curation, Formal analysis, Writing—original draft, Writing—review and editing; Guijun Shang, Methodology, Writing—original draft, Writing—review and editing; Hongtao Yu, Conceptualization, Formal analysis, Supervision, Funding acquisition, Validation, Writing—original draft, Writing—review and editing; Xiao-chen Bai, Conceptualization, Data curation, Formal analysis, Supervision, Funding acquisition, Validation, Writing—original draft, Writing—review and editing

### Author ORCIDs

Emiko Uchikawa  https://orcid.org/0000-0003-0442-767X
Eunhee Choi  https://orcid.org/0000-0003-3286-6477
Guijun Shang  http://orcid.org/0000-0002-0187-7934
Hongtao Yu  https://orcid.org/0000-0002-8861-049X
Xiao-chen Bai  https://orcid.org/0000-0002-4234-5686

### Decision letter and Author response

Decision letter https://doi.org/10.7554/eLife.48630.029
Author response https://doi.org/10.7554/eLife.48630.030

## Additional files

### Supplementary files

• Source data 1. All the required soure data.
DOI: https://doi.org/10.7554/eLife.48630.018
• Transparent reporting form
DOI: https://doi.org/10.7554/eLife.48630.019

### Data availability

Cryo-EM maps and the corresponding bulit models of insulin receptor/insulin complex have been deposited in EMDB and PDB under the accession codes EMD-20522/EMD-20523 and 6PXV/6PXW, respectively. All data generated or analysed during this study are included in the manuscript and supporting files. Source data files have been provided for Figure2, Figure 3, Figure 4, Figure 1—figure supplement 1, Figure 3—figure supplement 2 and Figure 3—figure supplement 3.

The following datasets were generated:

| | Database and |
|---|---|

| Author(s) | Year | Dataset title | Dataset URL | Identifier |
|---|---|---|---|---|
| Uchikawa E, Choi E, Shang G, Yu H, Bai X-C | 2019 | Cryo-EM structure of full-length insulin receptor bound to 4 insulin. 3D refinement was focused on the extracellular region | https://www.ebi.ac.uk/pdbe/entry/emdb/EMD-20522 | Electron Microscopy Data Bank, EMD-20522 |
| Uchikawa E, Choi E, Shang G, Yu H, Bai X-C | 2019 | Cryo-EM structure of full-length insulin receptor bound to 4 insulin. 3D refinement was focused on the top part of the receptor complex | https://www.ebi.ac.uk/pdbe/entry/emdb/EMD-20523 | Electron Microscopy Data Bank, EMD-20523 |
| Uchikawa E, Choi E, Shang G, Yu H, Bai X-C | 2019 | Cryo-EM maps | http://www.rcsb.org/structure/6PXW | Protein Data Bank, 6PXW |
| Uchikawa E, Choi E, Shang G, Yu H, Bai X-C | 2019 | Cryo-EM maps | http://www.rcsb.org/structure/6PXV | Protein Data Bank, 6PXV |

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
