## [Decision Letter]

Thank you for submitting your article "Activation mechanism of the insulin receptor revealed by cryo-EM structure of the fully liganded receptor-ligand complex" for consideration by *eLife*. Your article has been reviewed by three peer reviewers, one of whom, Nikolaus Grigorieff, is a member of our Board of Reviewing Editors, and the evaluation has been overseen by a Reviewing Editor and Philip Cole as the Senior Editor.

The reviewers have discussed the reviews with one another and the Reviewing Editor has drafted this decision to help you prepare a revised submission.

Summary:

Uchikawa et al. describe cryo-EM studies of the full-length insulin receptor (in DDM micelles). By focusing on the extracellular region alone, their structure does provide some new insights that are important for the field. It has long been proposed that each IR has two binding sites for insulin, and here – with excess/saturating insulin – the authors observe that there two sites on each receptor occupied. This is the first structural view of the 'second' binding site – which was previously identified only through alanine scanning and other biochemical techniques. However, revisions are required to ensure that the 2-fold symmetry is not an image processing artifact, and certain statements should be toned down and qualified, as outlined in the points below.

Essential revisions:

1) The cryo-EM work looks solid. However, since one of the main points of the manuscript is the number of insulin molecules bound to the receptor, it is important to make sure that there are no image processing artifacts. Insulin has a molecular mass of about 5 kDa, small for cryo-EM purposes. This means detecting the presence of insulin at any of the binding sites might not be completely reliable. The authors rely in their analysis on the 3D classification without symmetry, concluding that their complex has 2-fold symmetry because the one good class they obtained does not display significant asymmetry. However, it is uncertain if asymmetric binding of the 5 kDa insulin would have been detected by this classification. It is a bit surprising that the authors apparently did not find even a minor class with sub-stoichiometrically bound insulin. This would have validated that classification can distinguish different binding site occupancies. The authors should double-check the classes and particle selections. If their classification procedure cannot make this distinction, the reconstruction shown by the authors could represent a mixture of occupied and unoccupied sites, averaging to the same overall ligand density in all sites and giving the false impression of 2-fold symmetry and four bound insulins. Ideally, to test if image processing can detect asymmetric binding, the authors could perhaps collect a dataset of receptor titrated with insulin to a level that is known to yield sub-stoichiometric binding, showing that they can detect this with their 3D classification.

2) The authors should also comment on the identity of the 926,144 particles that were initially picked but later discarded. The current analysis assumes that the activated state of the receptor is represented by a single, well-ordered conformation. Is it possible that there are other conformations hidden among the discarded particles that are more disordered and therefore did not yield "good" class averages? How can the authors exclude this possibility?

3) The horizontal part of the "T" (and site 1 occupancy) seems to be precisely the same as that published in previous studies – there is nothing new there. The arguments presented in the fourth paragraph of the subsection “Overall structure of full-length human IR bound to insulin” supporting the idea that the structure described here is *the* 'active' form (based largely on TM domain proximity in Figure 1—figure supplement 4B) are not convincing, though – and this presumption seems to limit the way in which the authors interpret their findings. The description of the structure and conformational changes – and associated mutation data – all seem fine and interesting. For site 1 binding, this echoes previous work. The description of the importance of site 2 binding seems more speculative. The clashes suggested in Figure 5 seem to make sense, but the model in Figure 7 (and Supplementary Video 1) are highly speculative, and how the 'two J-shaped protomers can spontaneously to form the T-shaped dimer through inter-protomer rotations' is not described at all. These aspects are just supposition. The authors should also try to put the potential activation mechanism in a better literature context – in light of the specific proposals made by De Meyts, Schafer, and others. At present, the authors present data that site 1 and site 2 are both important, but they do not present any data to suggest that 4:2 (Ins:IR) binding is important for activation (rather than 3:2 or even 2:2). Please soften the text where required and discuss these points.

4) Did the authors analyze insulin binding to the protein used for their EM studies? A crucial question – related to their discussion in the subsection “A tripartite interface between insulin 1 and site 1 stabilizes the active IR dimer” – is whether IR in these micelles behaves like IR in membranes or like the isolated extracellular region. Does this protein prep display negative cooperativity? Or is this complex one step beyond the 3:2 (Ins:IR) complex that De Meyt's proposed for the intact receptor in cells? Assessment of ligand binding characteristics – in the DDM micelles as used for EM (not in cells) – is essential for putting this structure in context.

5) The results presented in Figure 6 seem superfluous and superficial, and lack key controls. They would best be omitted. It is frequently the case that mutations in RTKs cause misfolding and aggregation in the ER – leading to constitutive activation that is not informative. The C647A mutation seems a likely candidate for such a mutation, since it involves a residue that is involved in maturation of the IR. No controls for this possibility (probability) are provided. Do the C647A variants make it to the cell surface? Do they even bind insulin? These are crucial controls that are currently absent – making the result uninterpretable at present.

6) Physiological Insulin concentrations sufficient to activate IR are in the 300 pM-1 nM range, and the authors' samples were prepared at >20 μM IR/Insulin concentrations-or concentrations >10,000 fold higher than physiological concentrations. The decreased but not abolished IR activity in IR variants with amino-acid substitutions at the second interface may also be explained by effects on the large and complex interdomain rearrangements that the author's and earlier IR/Ins structures show occur when ligand binds. Please comment and discuss.

7) A related structural study to this work was also recently deposited by Coskun's group to bioRxiv, adding confidence to the structural findings. Please comment and discuss.

---

## [Author Response]

Essential revisions:1) The cryo-EM work looks solid. However, since one of the main points of the manuscript is the number of insulin molecules bound to the receptor, it is important to make sure that there are no image processing artifacts. Insulin has a molecular mass of about 5 kDa, small for cryo-EM purposes. This means detecting the presence of insulin at any of the binding sites might not be completely reliable. The authors rely in their analysis on the 3D classification without symmetry, concluding that their complex has 2-fold symmetry because the one good class they obtained does not display significant asymmetry. However, it is uncertain if asymmetric binding of the 5 kDa insulin would have been detected by this classification. It is a bit surprising that the authors apparently did not find even a minor class with sub-stoichiometrically bound insulin. This would have validated that classification can distinguish different binding site occupancies. The authors should double-check the classes and particle selections. If their classification procedure cannot make this distinction, the reconstruction shown by the authors could represent a mixture of occupied and unoccupied sites, averaging to the same overall ligand density in all sites and giving the false impression of 2-fold symmetry and four bound insulins. Ideally, to test if image processing can detect asymmetric binding, the authors could perhaps collect a dataset of receptor titrated with insulin to a level that is known to yield sub-stoichiometric binding, showing that they can detect this with their 3D classification.

We greatly appreciate this constructive comment. We completely agree with the reviewers that the initial 3D classification with coarse angular sampling may not be good enough to separate the particles with different numbers of bound ligands, due to the fact that insulin is much smaller than IR. To address this issue, we performed symmetry expansion in combination with focused classification on insulin with signal subtraction of the receptor (a similar approach has been described in our previous works, Bai et al., 2015; Zhou et al., 2015). As the insulin molecules are too small to be aligned on their own, during focused classification, the orientations of the insulins were kept the same as those determined by the consensus C2 refinement of the entire complex. As a result, we can identify two major classes having different stoichiometries for insulin binding. In particular, 63% of the particles have 4 insulins bound (two each at sites 1 and 2), while 33% of the particles have 3 insulins bound (two at site 1; one at site 2). We performed another 3D refinement with C1 symmetry applied for the particles with only 3 insulin bound, and found that there is only a subtle difference between the structures of IR with 3 or 4 insulins bound, which indicates that binding of insulins to both sites 2 and 2’ is not required for the maintenance of the T-shaped IR dimer. This notion is consistent with our structural analysis that the T-shaped dimer is mainly stabilized by the tripartite interface formed between insulin 1 and several domains of IR. Importantly, the vast majority (96%) of the T-shaped particles have at least one insulin bound to site 2 or 2’, suggesting that insulin binding to site 2 plays an important role in promoting the conformational change of IR to form the active T shape. We have added a new supplementary figure (Figure 1—figure supplement 5) and a new paragraph in the main text to describe this new 3D classification result in the revised manuscript.

We thank the reviewers for the great suggestion of collecting cryo-EM data on a series of IR samples with different insulin concentrations. In this way, we may be able to solve the structures for intermediate conformational states of IR that have one or two insulins bound. This will be our next major goal. In parallel, we also plan to use the site 2 mutations to trap IR in these intermediate states. This is clearly beyond the scope of this particular paper, however. We hope the reviewers would agree.

2) The authors should also comment on the identity of the 926,144 particles that were initially picked but later discarded. The current analysis assumes that the activated state of the receptor is represented by a single, well-ordered conformation. Is it possible that there are other conformations hidden among the discarded particles that are more disordered and therefore did not yield "good" class averages? How can the authors exclude this possibility?

We thank the reviewers for this critical comment. Among the 926,144 particles, 868,657 particles were discarded after the 2D classification. To explain the criteria used to discard particles, we show all the class averages in Author response image 1, with some representative classes not chosen for further processing highlighted in red. The majority of 868,657 discarded particles were empty detergent micelles and mis-picked junk particles. To prevent bias in selecting class averages, we chose nearly all remaining classes for subsequent 3D classification. On the other hand, the extreme hydrophobic air-water interface on the cryo-EM grids can potentially damage our protein complex. This type of protein damage can be more severe to cell receptors with an elongated shape, such as IR. This could explain why some of the 3D classes show similar shapes to the “good” classes, but were only resolved at low resolution. Discarding particles from such “bad” classes may remove damaged particles from the dataset, thereby improving the quality for the final 3D reconstruction. As excess insulin was added to the final sample to saturate IR binding, we believe that our cryo-EM sample used in this work is relatively homogenous, which facilitates the generation of high-resolution cryo-EM maps.

**Author response image 1. respfig1:** EM analysis of IR. Some representative classes that are not selected for further processing are highlighted in red box.

3) The horizontal part of the "T" (and site 1 occupancy) seems to be precisely the same as that published in previous studies – there is nothing new there. The arguments presented in the fourth paragraph of the subsection “Overall structure of full-length human IR bound to insulin” supporting the idea that the structure described here is the 'active' form (based largely on TM domain proximity in Figure 1—figure supplement 4B) are not convincing, though – and this presumption seems to limit the way in which the authors interpret their findings. The description of the structure and conformational changes – and associated mutation data – all seem fine and interesting. For site 1 binding, this echoes previous work. The description of the importance of site 2 binding seems more speculative. The clashes suggested in Figure 5 seem to make sense, but the model in Figure 7 (and Supplementary Video 1) are highly speculative, and how the 'two J-shaped protomers can spontaneously to form the T-shaped dimer through inter-protomer rotations' is not described at all. These aspects are just supposition. The authors should also try to put the potential activation mechanism in a better literature context – in light of the specific proposals made by De Meyts, Schafer, and others. At present, the authors present data that site 1 and site 2 are both important, but they do not present any data to suggest that 4:2 (Ins:IR) binding is important for activation (rather than 3:2 or even 2:2). Please soften the text where required and discuss these points.

We greatly appreciate this reviewers’ critical comment. We completely agree with the reviewers that we do not know for certain that the T-shaped dimer is the only active form of IR. It is likely that IR with different numbers of insulin bound may adopt different active conformations. Indeed, our cell-based assays show that the IR site 2 mutant exhibits approximately 50% activity as compared to IR WT, not completely inactive. We have changed the text from “the active form” to “one of its active forms”.

We agree with the reviewers that, due to the lack of structures of intermediate states of IR, we do not know how the two J-shaped protomers undergo structural rearrangements to form the T-shaped dimer. We have removed the speculative text, and stated that “Based on the current models, it is still unclear how the two “J”-shaped protomers form the “T”-shaped dimer.” in the revised manuscript. We have removed Supplementary Video 1 from supplementary materials and modified the working model in Figure 6 in the revised manuscript.

Based on the current results, the dual roles of insulin 1 binding for IR activation have been well defined. Insulin binding to site 1 is required to disrupt the auto-inhibited apo-state, and it subsequently stabilizes the active dimer by simultaneously contacting multiple domains of both IR protomers. This activation mechanism involving insulin 1 binding strongly supports the long-known “cross-linking” model first proposed by Schafer in 1994. On the other hand, the functional role of insulin 2 is not completely understood. Based on our new structural model, we propose that insulin binding to site 2 of the apo-state of IR may help to disrupt the auto-inhibited conformation, which promotes the formation of the active state of IR. Moreover, a characteristic feature of insulin binding to IR is negative cooperativity, which was firstly observed by De Meyts in 1973. Interestingly, our dose-response activity assays show that the IR site 2 mutant exhibits stronger negative cooperativity than IR WT, suggesting that insulin binding to site 2 reduces the negative cooperativity between site 1 and site 1’. Thus, we propose that following the binding of the first insulin to site 1, insulin binding to site 2 may facilitate the binding of the second insulin 1 to site 1’, thereby driving the formation of the symmetrical T-shaped IR dimer and full IR activation. Consistent with this hypothesis, our new focused 3D classification results suggest that binding of two site-1 insulins and the forming the T-shaped IR dimer require the binding of at least one site-2 insulin. Solving the structures of IR site 1 and 2 mutants in the presence of insulin is needed to further test this hypothesis.

We have softened our text and added discussion about these points in the revised manuscript.

4) Did the authors analyze insulin binding to the protein used for their EM studies? A crucial question – related to their discussion in the subsection “A tripartite interface between insulin 1 and site 1 stabilizes the active IR dimer” – is whether IR in these micelles behaves like IR in membranes or like the isolated extracellular region. Does this protein prep display negative cooperativity? Or is this complex one step beyond the 3:2 (Ins:IR) complex that De Meyt's proposed for the intact receptor in cells? Assessment of ligand binding characteristics – in the DDM micelles as used for EM (not in cells) – is essential for putting this structure in context.

We greatly appreciate the reviewers’ comment. As suggested, we have tested insulin binding to the same full-length IR sample used for cryo-EM analysis. Our binding results showed that recombinant IR isolated with the detergent DDM binds insulin with high affinity and displays negative cooperativity, comparable to our cell-based insulin-binding results. This finding suggests that DDM does not change insulin-binding behaviors. We have incorporated the new results into Figure 1—figure supplement 1.

In addition, the cryo-EM structure of IR extracellular domain (ECD) bound with 4 insulins recently solved by another group displays a nearly identical architecture to our DDM-isolated full-length IR/insulin complexes.

5) The results presented in Figure 6 seem superfluous and superficial, and lack key controls. They would best be omitted. It is frequently the case that mutations in RTKs cause misfolding and aggregation in the ER – leading to constitutive activation that is not informative. The C647A mutation seems a likely candidate for such a mutation, since it involves a residue that is involved in maturation of the IR. No controls for this possibility (probability) are provided. Do the C647A variants make it to the cell surface? Do they even bind insulin? These are crucial controls that are currently absent – making the result uninterpretable at present.

We agree with the reviewers that this result is peripheral to our main conclusions. We have removed the data and discussion on the C647A mutation in the revised manuscript.

6) Physiological Insulin concentrations sufficient to activate IR are in the 300 pM-1 nM range, and the authors' samples were prepared at >20 μM IR/Insulin concentrations-or concentrations >10,000 fold higher than physiological concentrations. The decreased but not abolished IR activity in IR variants with amino-acid substitutions at the second interface may also be explained by effects on the large and complex interdomain rearrangements that the author's and earlier IR/Ins structures show occur when ligand binds. Please comment and discuss.

We thank the reviewers for this critical comment, and completely understand their concern. To ease this concern, we would like to point out that all our cell-based IR activation and insulin binding assays for wild-type IR, IR site 1 mutants and IR site 2 mutants were done at insulin concentrations ranging from 0.8 nM to 80 nM, which is close to the blood insulin concentrations, as mentioned by the reviewer. The results show that insulin binding at site 2 is important for full IR activation at physiological insulin concentration. In fact, IR site 2 mutants exhibited even greater insulin binding deficiency at low insulin concentrations (0.8 nM), as compared to that at higher insulin concentrations, confirming that insulin binding at site 2 occurs at physiologically relevant insulin concentrations.

The structure of FnIII-1 of insulin-bound IR can be superimposed perfectly onto that of apo-IR, suggesting that these surface residues at the second interface in FnIII-1 domain are unlikely to play any roles in driving the interdomain rearrangements during IR activation. Furthermore, previous mutagenesis analyses of insulin have identified the insulin residues responsible for binding to site 2. These insulin residues are located at the new IR-insulin interface at site 2 in our structure. Thus, mutations of interface residues of both IR (by us in this study) and insulin (reported previously by others) confirm the functional importance of the new site 2 in insulin-dependent activation of IR, and completely exclude the possibility that the site 2 identified in this work is a structural artifact.

We agree with the reviewers that the IR site 2 mutations only decreased, but did not abolish, IR activity. One possible explanation is that, at low insulin concentrations, IR with partial insulin occupancy (e.g. one insulin at site 1) can achieve asymmetric conformations and partial activation. Increased insulin occupancy at site 2 promotes the formation of the T-shaped symmetric conformation and full activation. It is tempting to speculate that the conformational differences of IR caused by different stoichiometries of bound insulins may provide a mechanism in defining insulin signaling specificity, which could potentially allow the same ligand to generate distinct signaling outputs from the same receptor. Future biophysical and in vivo experiments, preferably those involving single-molecule techniques, are required to define the intermediate steps of IR activation.

We have toned down our conclusions and added discussion about these points in the revised manuscript.

7) A related structural study to this work was also recently deposited by Coskun's group to bioRxiv, adding confidence to the structural findings. Please comment and discuss.

We thank the reviewers for this good point. Indeed, this cryo-EM structure of IR ECD bound with 4 insulins shows nearly identical architecture to our full-length IR/insulin complexes, supporting our new finding. Nevertheless, our cryo-EM map was resolved at much higher resolution, which provides more structural details. Our functional studies are also more thorough. We have cited and discussed this preprint in the revised manuscript.